# Exploring Deeper! Segment Anything Model with Depth Perception for Camouflaged Object Detection

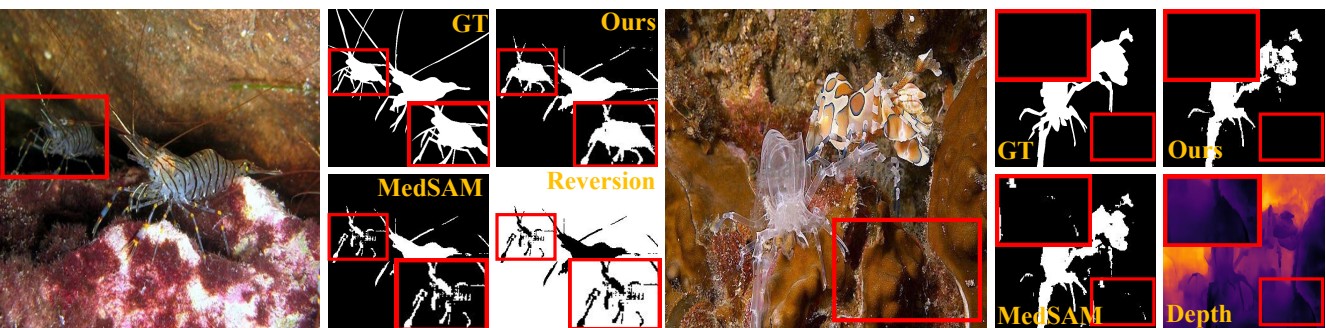

**Figure 1: Visual comparison between the results of our proposed DSAM and the MedSAM [25]. The left part reflects the role of Finer Module. The missing part is re-mined by mask reversion operation and segmented with the help of depth features. The right part embodies the role of Prompt-Deeper Module, which removes the segmented regions skillfully that show anomalies in the depth maps.**

## ABSTRACT

This paper introduces a new Segment Anything Model with Depth Perception (DSAM) for Camouflaged Object Detection (COD). DSAM exploits the zero-shot capability of SAM to realize precise segmentation in the RGB-D domain. It consists of the Prompt-Deeper Module and the Finer Module. The Prompt-Deeper Module utilizes knowledge distillation and the Bias Correction Module to achieve the interaction between RGB features and depth features, especially using depth features to correct erroneous parts in RGB features. Then, the interacted features are combined with the box prompt in SAM to create a prompt with depth perception. The Finer Module explores the possibility of accurately segmenting highly camouflaged targets from a depth perspective. It uncovers depth cues in areas missed by SAM through mask reversion, self-filtering, and self-attention operations, compensating for its defects in the COD domain. DSAM represents the first step towards the SAM-based RGB-D COD model. It maximizes the utilization of depth features while synergizing with RGB features to achieve multimodal complementarity, thereby overcoming the segmentation limitations of SAM and improving its accuracy in COD. Experimental results on COD benchmarks demonstrate that DSAM achieves excellent segmentation performance and reaches the state-of-the-art (SOTA) on COD benchmarks with less consumption of training resources.

## CCS CONCEPTS

• **Computing methodologies** → **Object detection**; *Image segmentation*; Computer vision.

## KEYWORDS

Camouflaged Object Detection, Segment Anything Model, RGB-D

## 1 INTRODUCTION

Segment Anything Model (SAM) [17] is a visual foundational model with robust zero-shot capabilities and high versatility. It can segment a wide range of objects through various forms of interactive prompts. However, for specific domains such as Camouflaged Object Detection (COD), SAM does not perform well due to the deceptive nature of objects in this domain, which differs significantly from the SA-1B dataset used during training. Some studies have to some extent overcome this issue and promoted the development of SAM in the field of COD. Chen *et al.* [3] utilized adapter technology to provide SAM with relevant information for COD tasks, enabling it to adapt to COD tasks. Ma *et al.* [25] took box prompts as interactive prompts to fine-tune the mask decoder, achieving progress in medical image domains and COD. However, SAM cannot effectively segment highly disguised areas. One reason is that current SAM-based COD work is limited to the RGB level, failing to obtain semantic and structural information from highly disguised components.

To solve this problem, this paper proposes a SAM-based RGB-D COD model, the Segment Anything Model with Depth Perception (DSAM) for Camouflaged Object Detection. Considering the large depth disparity between the target and the background, DSAM captures the semantic and structural features of the target from the RGB-D perspective, compensating for the information loss caused by heavy camouflage in the RGB domain. Meanwhile, DSAM fully exploits the complementarity among multiple modalities to improve segmentation performance. Specifically, two modules are proposed:

the Prompt-Deeper Module (PDM) and the Finer Module (FM). Considering that depth maps may introduce some noise, some methods are employed to suppress noise when designing the modules. By using PVT, PDM initially extracts image features to derive student embedding while using a frozen-parameter image encoder to extract depth maps features to generate teacher embedding. Through knowledge distillation and the Bias Correction Module (BCM), the student embeds depth information from the teacher and interacts with its RGB information, achieving mutual information complementation between the two modalities. Additionally, distillation is utilized to focus on learning the content of features instead of changing the model size. Moreover, student embedding is combined with the original box prompt to integrate depth information into the prompt, and then a box-shaped interactive prompt with depth perception capability emerges. Compared to directly integrating depth information, our method greatly reduces the impact of noise from depth maps , as illustrated in the right part of Fig. 1. This transformation of the original box prompt into a depth-inclusive box prompt improves the efficiency of interactive prompts.

Furthermore, there are instances of omissions in the SAM segmentation prediction map, as demonstrated in the left part of Fig. 1. To solve this problem, the FM employs mask reversion operations to focus on the omitted parts of the SAM segmentation, and the module adopts a two-stream to one-stream architecture. In the two-stream structure, different strategies are employed for depth embedding. Specifically, in stream 1, self-guided filtering [22] is applied to focus on target edges through self-selection. In stream 2, agent attention [9] is utilized to concentrate on target regions and establish long-distance modeling. Then, through joint mining, the two streams are combined, and further exploration is conducted using agent attention in stream 2. Finally, PDM and FM are ingeniously combined with SAM to propose DSAM. DSAM excavates deep information and integrates a novel prompt endowed with depth perception capabilities. It further segments overlooked portions in SAM from the perspective of depth information, thereby strategically enhancing SAM's evaluation metrics in COD. Our contributions are summarized as follows:

- DSAM, a variant of SAM with depth information tailored for the COD domain is proposed. To the best of our knowledge, DSAM is the first SAM-based RGB-D COD model. The model explores the interaction between depth information and RGB information in the COD domain under the SAM framework, where this interaction serves a modal complementary role.
- Two modules in DSAM are proposed, namely PDM and FM. PDM achieves mutual complementation of two modalities by interacting through deep features and RGB features, leading to a novel prompt endowed with depth perception capabilities. By mining the depth cues of the overlooked segments in SAM segmentation, FM compensates for the original prediction results of SAM, thereby improving accuracy.
- DSAM is compared with 17 existing methods on COD benchmark datasets. In scenarios with limited training resources, DSAM outperforms existing state-of-the-art (SOTA) methods. Meanwhile, to comprehensively demonstrate the performance of DSAM, it is compared with the RGB-D model with source-free depth, and DSAM exhibits superior metrics.

## 2 RELATED WORK

### 2.1 Camouflaged Object Detection

In nature, animals employ camouflage as a protective mechanism to avoid predators [4]. Because of the highly sophisticated disguise techniques, COD has always been a challenging downstream task. With the advancement of deep neural networks, various breakthroughs have promoted the development of COD. Pang *et al.* [28] employed a zoom strategy to explore mixed-scale semantics, mining cues between candidate objects and background context. Inspired by human attention mechanisms, Jia *et al.* [16] proposed an iterative refinement framework, involving three stages: segment, magnify, and reiterate, which yields promising results. Some work designed networks based on patterns of animal hunting (SINetV2 [7], SLSR [24], PFNet [27]). Yang *et al.* [37] combined a probabilistic representation model with transformers and utilized uncertainty to guide transformer inference for disguised object detection. In addition to the efforts made from the RGB perspective, some studies delved into the relationship between image depth from the RGB-D perspective and transformed the RGB COD task into an RGB-D COD task. Xiang *et al.* [36] first proposed the potential contribution of depth to COD, but a dedicated RGB-D dataset was lacking. The depth maps used in this method contained considerable noise, so it only served as an auxiliary branch rather than being directly integrated. In recent studies, Bi *et al.* [2] introduced a dynamic allocation mechanism during the fusion process of depth maps, and they employed a criterion termed as depth alignment index. Liu *et al.* [23] proposed a multi-scale fusion approach to suppress the interference of inaccurate depth maps on camouflage target detection. In the aforementioned approaches based on RGB-D COD, the direct fusion of depth maps is commonly used, which inevitably introduces certain levels of noise. In this paper, knowledge distillation is employed to learn depth information from depth maps, and an indirect approach is utilized to interact RGB information with depth information, aiming to reduce the noise introduced into depth maps.

### 2.2 Camouflaged Object Detection based SAM

The proposal of SAM brings a highly versatile and strongly generalizable visual foundational model to the field of computer vision. However, when SAM is applied to specific downstream tasks such as COD and medical image segmentation, the issue of low segmentation accuracy cannot be ignored. This issue is due to the significant domain gap between the dataset used for SAM training and the specific downstream domain dataset. Also, directly applying SAM lacks semantic guidance in that domain. The literature [14] investigated SAM's effectiveness in concealed scenarios and concluded that SAM is inexperienced in such contexts. Tang *et al.* [32] directly investigated COD and pointed out that SAM has limited performance on the COD task. Recently, researchers are attempting to fill this gap. In weakly-supervised concealed object segmentation, He *et al.* [11] utilized sparse annotations provided by SAM as cues for mask segmentation in model training. Chen *et al.* [3] integrated domain-specific information into the model using adapters. However, the aforementioned COD models based on SAM mainly address information at the RGB level, which complicates the segmentation of highly deceptive regions. In this paper, the

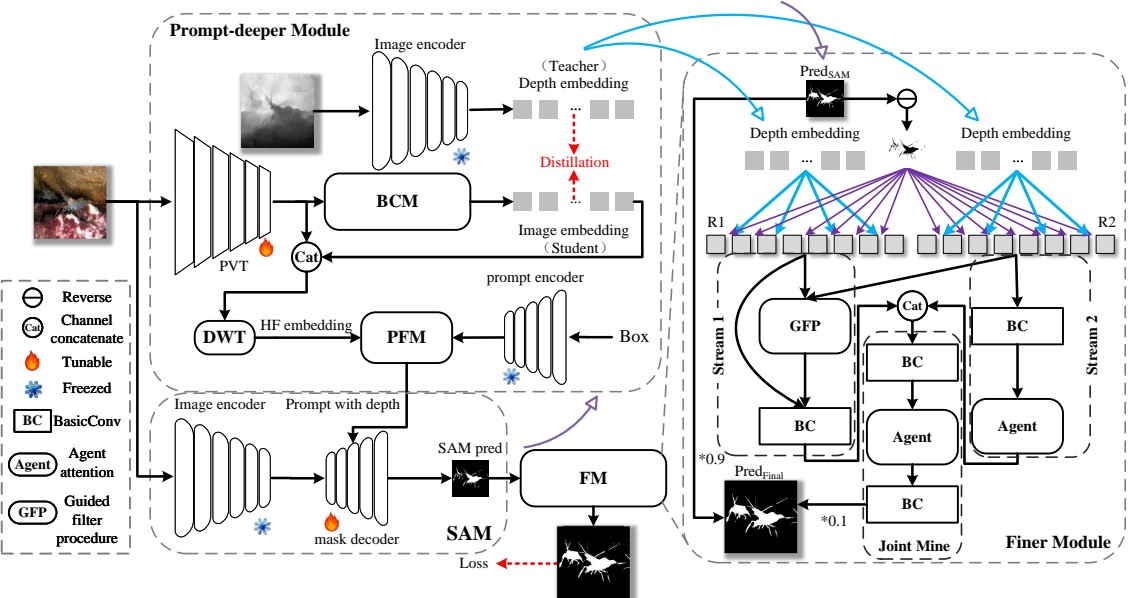

**Figure 2: Overview of our DSAM framework. It mainly includes three parts: SAM, the Prompt Deeper Module (PDM), and the Finer Module (FM). PDM consists of the Bias Correction Module (BCM), the Prompt Fuse Module (PFM) and Discrete Wavelet Transform (DWT). Regarding modules in SAM and PVT, the parameters in the module with snowflake are fixed, while the parameters in the module with spark can be optimized by training. In order to enhance the comprehensibility of the figure, we annotate the sources of the inputs to FM with arrows. Purple arrows indicate that $Pred_{SAM}$ originates from the output of SAM. Blue arrows represent depth embedding originating from the intermediate embedding of the PDM.**

formidable camouflage characteristics in the field of COD are addressed from the RGB-D perspective to avoid superb camouflage techniques in the RGB domain. By exploiting depth information, the development of SAM in the COD domain is further promoted.

## 3 METHODOLOGY
### 3.1 Overall Architecture
Fig. 2 illustrates the overall architecture of the proposed DSAM. The entire model consists of three components: SAM, PDM, and FM, where the latter two are pivotal. Overall, two improvements are made to SAM: the box prompt is upgraded to a box prompt with depth perception capability, and the prediction maps of SAM segmentation are refined. First, the image is fed into SAM. In SAM, this study employs a strategy wherein the parameters of the image encoder and prompt encoder are kept static, while those of the mask decoder undergo fine-tuning [25]. This strategy retains the powerful feature extraction capability of SAM, reduces computational complexity, and adapts to downstream tasks. Then, the box prompt is refined to obtain a box prompt enriched with depth perception by PDM through information interaction between RGB and depth. Compared to existing fusion-based methods, less noise is generated during the depth acquisition via PDM, resulting in enhanced utilization of depth information. Then, SAM generates prediction maps by using the new box prompt. To further enhance the accuracy of SAM in COD, our strategy targets the often neglected segments within SAM segmentation. To re-segment these regions, FM is designed to complement the segmentation of the predicted image by extracting

deep features from the overlooked segments. By adequately integrating and exploiting the depth information and RGB information through PDM and FM, the segmentation accuracy of SAM in COD is further enhanced.

### 3.2 Prompt Deeper Module
PDM is designed to explore the reasonable interaction between RGB features and depth features while suppressing noise in the depth map. In PDM, images ($I$) are processed using PVT [34] ($T_{image}$) and Bias Correction Module (BCM) to extract features for obtaining student embedding ($Em_s$), while depth maps ($D$) are processed a frozen-parameter image encoder ($T_{depth}$) to extract features for obtaining teacher embedding ($Em_t$). The adoption of PVT for obtaining student embedding, instead of relying on a frozen image encoder, serves dual purposes: it provides the student embedding a broader scope for learning adjustments and PVT demonstrates superior feature extraction capabilities compared to image encoder [34]. After passing through knowledge distillation and BCM, student embedding learns depth information from teacher embedding and interacts with RGB information to achieve complementary effects. Knowledge distillation is usually used to distill information from large models, but our purpose in using it is to perform learning between different features. Beyond this, the depth maps are not obtained through depth sensors, so they inevitably contain noise. If they are directly fused, the adverse noise effects introduced by the depth map outweigh the beneficial depth information. Therefore, this study does not adopt direct fusion but instead employs a completely new approach: using knowledge distillation with depth

embedding serving as the teacher and image embedding as the student. The student learns depth features from the teacher through knowledge distillation. This to some extent reduces the noise introduced by directly fusing deep features. The process of knowledge distillation and BCM is expressed as follows:

$$Em_t = T_{depth}(D), \qquad (1)$$

$$Em_i = T_{image}(I), \qquad (2)$$

$$Em_s = Down\left(CP(P(Em_i))\right), \qquad (3)$$

$$Loss_{KD} = L_{CWD}(Em_t, Em_s), \qquad (4)$$

where $P$ represents the projection with a constant number of channels. $CP$ denotes channel projection, which involves the operations of enlarging and shrinking the number of channels by dilated convolution. $Down$ denotes the projection with decreasing number of channels. $L_{CWD}$ represents channel-wise knowledge distillation [29]. The residual and sampling operations are omitted from the above formulas for brevity. The frameworks of BCM and PFM are shown in Fig. 3. Afterwards, the student embedding extracts high-frequency information corresponding to edge details through DWT [10], followed by fusion using the PFM and box prompt, leading to a new box prompt enriched with depth information. The following are the formulas of integration:

$$HF = DWT\left(cat(Em_i, Em_s)\right) \qquad (5)$$

$$Promtp_{depth} = PFM(Em_b, HF) = DC\left(cat\left(DCs(Em_b), HF\right)\right) \qquad (6)$$

where $cat(\cdot)$ denotes the join operation by channel dimension, $Em_b$ represents the prompt embedding obtained from box prompt after prompt embedding, $HF$ denotes the part of high frequency, $DC$ represents dilated convolution and $DCs$ represents being composed of convolution. This leads to the formation of a box prompt embedded with depth information.

### 3.3 Finer Module

SAM generates prediction by using a box prompt with depth information. Furthermore, FM is proposed to delve into deep information and segment the parts ignored by SAM to enhance the original prediction results. Our focal is on the overlooked portions of SAM, and there are two reasons behind this: firstly, owing to SAM's proficient zero-shot capability, it can adequately segment camouflaged targets. By excluding these adequately segmented regions, our network can concentrate more on exploring neglected areas, yielding a higher efficiency; secondly, our network can reduce the size of the target area with alleviated adverse effects of noise of depth maps. By applying the mask reversion operation [7], which involves reversing the original prediction map generated by SAM ($Pred_{SAM}$), the issue of omitted regions can be addressed. FM employs a two-stream to one-stream architecture. The depth embedding is divided into eight segments along the channel dimension, and the inverted prediction map within it is nested, thereby creating a feature with both depth information and occluded positions. To comprehensively explore depth information, both input channels are composed of nested depth embedding, with each channel emphasizing different measures. This process can be expressed as follows:

$$R_1 = \Gamma(Em_t, pred_{SAM}) \qquad (7)$$

$$R_2 = \Gamma(Em_t, pred_{SAM}) \qquad (8)$$

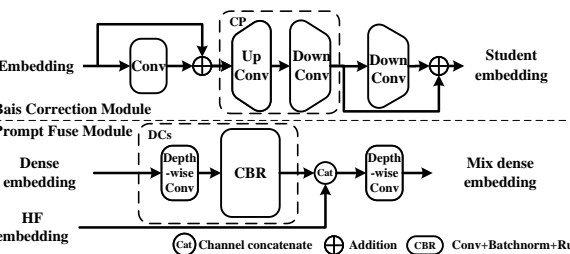

**Figure 3: Structure diagram of Bias Correction Module and Prompt Fuse Module.** $CP$ **represents channel projection, including** $Up\ conv$ **and** $down\ conv.$ $Up\ conv$ **refers to convolution that increase the number of channels, while** $down\ conv$ **refers to convolution that decrease the number of channels.** $DCs$ **denotes the composition of convolution.**

where $\Gamma$ represents the operation of mask reversion and nesting the inverted prediction map. $R_1$ and $R_2$ denote the original features in stream 1 and stream 2, respectively. Then, the features are fed into a dual-stream structure. In stream 1, $R_1$ is processed by the guided filter procedure (GFP) [22], and it is filtered to maintain the accuracy of the target edges. To comprehensively explore deep information, agent attention [9] (agent) is directly applied in stream 2 for long-range modeling. Finally, the dual-stream structure is concatenated into a single stream, followed by another round of agent attention to extract useful information for an optimized prediction map.

$$Pred_{FM} = JM\left(BC\left(GFP(R_1)\right) + agent\left(BC(R_2)\right)\right), \qquad (9)$$

where $JM$ denotes joint mining, including BC and agent attention. To integrate the re-segmented maps of FM with the original maps generated by SAM, a proportional summation approach is employed. Then, the final predictive map of DSAM is obtained. The formula is as follows:

$$Pred_{final} = (1-\alpha)Pred_{FM} + \alpha Pred_{SAM}, \qquad (10)$$

where $\alpha$ is a hyper-parameter to balance the original prediction map of SAM and the prediction map of FM (see more details in Sec. 4.4).

### 3.4 Framework Optimization

In this section, the composition of the entire network's loss function is introduced. While retaining SAM's original loss function, this study incorporates the distillation loss function from the PDM module ($Loss_{KD}$), and the channel-wise knowledge distillation loss [29] is utilized as the loss function for PDM. The channel-wise distillation loss utilizes softly aligned activation of corresponding channels, fully leveraging the knowledge within each channel. SAM employs the DiceCELoss ($Loss_{SAM}$) as its loss function, which computes the weighted sum of both the dice loss and cross-entropy loss. To simultaneously optimize the corresponding parts of the network for the two loss functions, this study adopts the approach of adding these two functions together in appropriate proportions, forming the loss function for DSAM. The specific formula is given below:

$$Loss = \beta Loss_{SAM} + (1-\beta)Loss_{KD}, \qquad (11)$$

where $\beta$ is a hyper-parameter to balance the original loss of SAM and the distillation loss of PDM (see more details in Sec. 4.4).

**Figure 4: Comparison of our DSAM and other methods in COD. Our approach enables a more comprehensive segmentation of camouflaged targets (row four and eight), while also exhibiting precise attention to detail (row one and three). DSAM achieves relatively satisfactory segmentation results in multi-objective scenarios as well (row nine to eleven). Better to zoom in.**

## 4 EXPERIMENTS

### 4.1 Dataset

Experiments are conducted on three widely recognized datasets, namely CAMO [18], COD10K [7], and NC4K [24]. CAMO comprises 1250 images, divided randomly into a training dataset of 1000 images and a testing dataset of 250 images. COD10K contains 5066 images, with 3040 assigned to the training dataset and 2026 to the testing dataset. NC4K contains a total of 4121 images. NC4K serves as the testing dataset for experiments to evaluate the generalization capability of DSAM. Following Fan *et al.* [7], our study adopts a dataset that is composed of the training datasets of COD10K and CAMO, which include 3040 images and 1000 images, respectively. The remaining images of COD10K and CAMO, and the entire NC4K dataset are used as the test dataset.

### 4.2 Experimental Setup

**Implementation Details.** DSAM is implemented using PyTorch, and the Adam optimizer with a learning rate of $1e^{-5}$ is utilized. To achieve optimal performance, the model is trained in 100 epochs, and the process is completed in $\sim$ 7.5 hours with a batch size of 8. In particular, an NVIDIA 3080TI GPU device with 12 GB video memory is utilized, which has relatively modest requirements compared to

other methods. All the input images are scaled to $1024 \times 1024$ pixels through bilinear interpolation. Additionally, the input image data are truncated and normalized to make the pixel values fall in the appropriate range while maintaining the relative distribution relationship of the data.

**Evaluation Metrics.** Six evaluation metrics are adopted, which are widely used and recognized in the field of COD, including structure measure ($S_\alpha$) [5], weighted F-measure($F_\beta^\omega$) [26], mean F-measure [1], mean enhanced-alignment measure ($E_\phi^m$) [6], max enhanced-alignment measure ($E_\phi^x$) and mean absolute error ($M$). Specifically, the structure measure measures the structural similarity between the predicted results and the actual segmented regions. The F-measure is the harmonic mean of precision and recall. The enhanced-alignment measure evaluates the prediction result by comparing the alignment relationship between the predicted value and the actual value. The mean absolute error (MAE) is the average absolute difference between predicted values and true values.

### 4.3 Comparisons

Here, COMPrompter is compared with SAM [17] and other existing COD algorithms, such as SINet [8], SLSR [24], $C^2$FNet [30],

| Dataset | Metric | SINet | SLSR | $C^2$FNet | UJSC | UGTR | PFNet | R-MGL | BSANet | OCENet | BGNet | ZoomNet | SINetv2 | SegMaR | DGNet | FSPNet | SAM | MedSAM | Ours |
|---|---|---|---|---|---|---|---|---|---|---|---|---|---|---|---|---|---|---|---|
| | | 2020 | 2021 | 2021 | 2021 | 2021 | 2021 | 2021 | 2022 | 2022 | 2022 | 2022 | 2022 | 2023 | 2023 | 2023 | 2023 | 2023 | - |
| | | [8] | [24] | [30] | [20] | [37] | [27] | [38] | [42] | [21] | [31] | [28] | [7] | [16] | [13] | [12] | [17] | [25] | |
| CAMO | $S_\alpha \uparrow$ | 0.751 | 0.787 | 0.796 | 0.800 | 0.784 | 0.782 | 0.775 | 0.794 | 0.802 | 0.812 | 0.820 | 0.820 | 0.815 | 0.839 | **0.856** | 0.684 | 0.820 | 0.832 |
| | $F_\beta^\omega \uparrow$ | 0.606 | 0.696 | 0.719 | 0.728 | 0.684 | 0.695 | 0.673 | 0.717 | 0.723 | 0.749 | 0.752 | 0.743 | 0.753 | 0.769 | **0.799** | 0.606 | 0.780 | 0.794 |
| | $F_\beta^m \uparrow$ | 0.675 | 0.744 | 0.762 | 0.772 | 0.735 | 0.746 | 0.726 | 0.763 | 0.766 | 0.789 | 0.794 | 0.782 | 0.795 | 0.806 | **0.830** | 0.680 | 0.814 | 0.821 |
| | $E_\phi^m \uparrow$ | 0.771 | 0.838 | 0.854 | 0.859 | 0.822 | 0.842 | 0.812 | 0.851 | 0.852 | 0.870 | 0.878 | 0.882 | 0.874 | 0.901 | 0.899 | 0.687 | 0.902 | **0.913** |
| | $E_\phi^x \uparrow$ | 0.831 | 0.854 | 0.864 | 0.873 | 0.851 | 0.855 | 0.842 | 0.867 | 0.865 | 0.882 | 0.892 | 0.895 | 0.884 | 0.915 | **0.928** | 0.689 | 0.913 | 0.920 |
| | $M \downarrow$ | 0.100 | 0.080 | 0.080 | 0.073 | 0.086 | 0.085 | 0.088 | 0.079 | 0.080 | 0.073 | 0.066 | 0.070 | 0.071 | 0.057 | **0.050** | 0.132 | 0.065 | 0.061 |
| COD10K | $S_\alpha \uparrow$ | 0.771 | 0.804 | 0.813 | 0.809 | 0.817 | 0.800 | 0.814 | 0.818 | 0.827 | 0.831 | 0.838 | 0.815 | 0.833 | 0.822 | **0.851** | 0.783 | 0.841 | 0.846 |
| | $F_\beta^\omega \uparrow$ | 0.551 | 0.673 | 0.686 | 0.684 | 0.666 | 0.660 | 0.666 | 0.699 | 0.707 | 0.722 | 0.729 | 0.680 | 0.724 | 0.693 | 0.735 | 0.701 | 0.751 | **0.760** |
| | $F_\beta^m \uparrow$ | 0.634 | 0.715 | 0.723 | 0.721 | 0.712 | 0.701 | 0.711 | 0.738 | 0.741 | 0.753 | 0.776 | 0.718 | 0.757 | 0.728 | 0.769 | 0.756 | 0.782 | **0.789** |
| | $E_\phi^m \uparrow$ | 0.806 | 0.880 | 0.890 | 0.884 | 0.853 | 0.877 | 0.852 | 0.891 | 0.894 | 0.901 | 0.888 | 0.887 | 0.899 | 0.896 | 0.895 | 0.798 | 0.917 | **0.921** |
| | $E_\phi^x \uparrow$ | 0.868 | 0.892 | 0.900 | 0.891 | 0.890 | 0.890 | 0.890 | 0.901 | 0.905 | 0.911 | 0.911 | 0.906 | 0.906 | 0.911 | 0.930 | 0.800 | 0.926 | **0.931** |
| | $M \downarrow$ | 0.051 | 0.037 | 0.036 | 0.035 | 0.036 | 0.040 | 0.035 | 0.034 | 0.033 | 0.033 | 0.029 | 0.037 | 0.034 | 0.033 | **0.026** | 0.049 | 0.033 | 0.033 |
| NC4K | $S_\alpha \uparrow$ | 0.808 | 0.840 | 0.838 | 0.842 | 0.839 | 0.829 | 0.833 | 0.841 | 0.853 | 0.851 | 0.853 | 0.847 | 0.841 | 0.857 | **0.879** | 0.767 | 0.866 | 0.871 |
| | $F_\beta^\omega \uparrow$ | 0.723 | 0.766 | 0.762 | 0.771 | 0.747 | 0.745 | 0.740 | 0.771 | 0.785 | 0.788 | 0.784 | 0.770 | 0.781 | 0.784 | 0.816 | 0.696 | 0.821 | **0.826** |
| | $F_\beta^m \uparrow$ | 0.769 | 0.804 | 0.795 | 0.806 | 0.787 | 0.784 | 0.782 | 0.808 | 0.818 | 0.820 | 0.818 | 0.805 | 0.820 | 0.814 | 0.843 | 0.752 | 0.845 | **0.847** |
| | $E_\phi^m \uparrow$ | 0.871 | 0.895 | 0.897 | 0.898 | 0.875 | 0.888 | 0.867 | 0.897 | 0.903 | 0.907 | 0.896 | 0.903 | 0.896 | 0.911 | 0.915 | 0.776 | 0.927 | **0.932** |
| | $E_\phi^x \uparrow$ | 0.883 | 0.907 | 0.904 | 0.907 | 0.899 | 0.898 | 0.893 | 0.907 | 0.913 | 0.916 | 0.912 | 0.914 | 0.907 | 0.922 | 0.937 | 0.778 | 0.937 | **0.940** |
| | $M \downarrow$ | 0.058 | 0.048 | 0.049 | 0.047 | 0.052 | 0.053 | 0.052 | 0.048 | 0.045 | 0.044 | 0.043 | 0.048 | 0.046 | 0.042 | **0.035** | 0.078 | 0.041 | 0.040 |

**Table 1: Quantitative results of RGB COD model on three different datasets of CAMO, COD10K, and NC4K with six metrics. The scores in bold are best. ↑ indicates the higher the score the better and ↓ indicates the lower the score the better.**

| Dataset | Metric | CDINet | DCF | CMINet | SPNet | DCMF | SPSN | PopNet | Ours |
|---|---|---|---|---|---|---|---|---|---|
| | | 2021 | 2021 | 2021 | 2021 | 2022 | 2022 | 2023 | - |
| | | [39] | [15] | [40] | [41] | [33] | [19] | [35] | |
| CAMO | $F_\beta^{mx} \uparrow$ | 0.638 | 0.724 | 0.798 | 0.807 | 0.737 | 0.782 | 0.821 | **0.834** |
| | $S_\alpha \uparrow$ | 0.732 | 0.749 | 0.782 | 0.783 | 0.728 | 0.773 | 0.806 | **0.832** |
| | $E_\phi^x \uparrow$ | 0.766 | 0.834 | 0.827 | 0.831 | 0.757 | 0.829 | 0.869 | **0.920** |
| | $M \downarrow$ | 0.100 | 0.089 | 0.087 | 0.083 | 0.115 | 0.084 | 0.073 | **0.061** |
| COD10K | $F_\beta^{mx} \uparrow$ | 0.610 | 0.685 | 0.768 | 0.776 | 0.679 | 0.727 | 0.789 | **0.807** |
| | $S_\alpha \uparrow$ | 0.778 | 0.766 | 0.811 | 0.808 | 0.748 | 0.789 | 0.827 | **0.846** |
| | $E_\phi^x \uparrow$ | 0.821 | 0.864 | 0.868 | 0.869 | 0.776 | 0.854 | 0.897 | **0.931** |
| | $M \downarrow$ | 0.044 | 0.040 | 0.039 | 0.037 | 0.063 | 0.042 | **0.031** | 0.033 |
| NC4K | $F_\beta^{mx} \uparrow$ | 0.697 | 0.765 | 0.832 | 0.828 | 0.782 | 0.803 | 0.852 | **0.862** |
| | $S_\alpha \uparrow$ | 0.793 | 0.791 | 0.839 | 0.825 | 0.794 | 0.813 | 0.852 | **0.871** |
| | $E_\phi^x \uparrow$ | 0.830 | 0.878 | 0.888 | 0.874 | 0.820 | 0.867 | 0.908 | **0.940** |
| | $M \downarrow$ | 0.067 | 0.061 | 0.053 | 0.054 | 0.077 | 0.059 | 0.043 | **0.040** |

**Table 2: Quantitative results of RGB-D COD model on three different datasets of CAMO, COD10K, and NC4K with four metrics. The scores in bold are best. ↑ indicates the higher the score the better, ↓ indicates the lower the score the better. $F_\beta^{mx}$ denotes max F-measure [1].**

UJSC [20], UGTR [37], PFNet [27], R-MGL [38], BSANet [42], OCENet [21], BGNet [31], ZoomNet [28], SINetV2 [7], SegMaR [16], DGNet [13], FSPNet [12], MedSAM [25], as shown in Tab. 1. Although MedSAM operates in the field of medical image processing, its approach is to enhance SAM universally instead of tailoring a network specifically for medicine. When applied to the COD task, the algorithm significantly enhances performance metrics. Therefore, MedSAM

is taken as one of the comparison algorithms. To comprehensively demonstrate the performance of DSAM, it is also compared with existing RGB-D SOD methods, such as PopNet [35], SPSN [19], DCMF [33], SPNet [41], CMINet [40], DCF [15], and CDINet [39], as shown in Tab. 2. The predictions of the competitors are either disclosed by the authors or generated by models retrained using open-source code.

**Quantitative Result.** Tab. 1 summarizes the quantitative results of our proposed method compared to 17 competitors across COD benchmark datasets under six evaluation metrics. The results indicate that DSAM has certain advantages. DSAM significantly outperforms DGNet in five metrics on COD10K and NC4K. For example, on COD10K, DSAM achieves improvements of 6.7% and 6.1% in $F_\beta^\omega$ and $F_\beta^m$, respectively. On NC4K, our network achieves improvements of 4.2% and 3.3% in $F_\beta^\omega$ and $F_\beta^m$, respectively. Compared to the recent FSPNet, DSAM achieves improvements in four metrics on COD10K and NC4K. Among these, the $F_\beta^m$ and $E_\phi^m$ metrics of DSAM exceed those of FSPNet by 2%, 2.6%, 0.4%, and 1.7%, respectively. Additionally, to evaluate the competitiveness of DSAM in RGB-D models, it is compared with other RGB-D SOD models with source-free depth on the COD dataset. The data for the RGB-D SOD model with source-free depth originates from PopNet [35]. As depicted in Tab. 2, DSAM outperforms the listed methods in the fundamental metrics. In terms of average values, the positive indicators of DSAM are improved by 3% on CAMO, 2.3% on COD10K, and 2% on NC4K. Regarding the maximum values, the most significant improvement is observed in indicator $E_\phi^x$, with an increase of 5.1%, 3.4%, and 3.2% across the three metrics, respectively.

| | CAMO | | | | | | COD10K | | | | | | NC4K | | | | | |
|---|---|---|---|---|---|---|---|---|---|---|---|---|---|---|---|---|---|---|
| **Ablation study** | | | | | | | | | | | | | | | | | | |
| Model | $S_\alpha$ | $F_\beta^\omega$ | $F_\beta^m$ | $E_\phi^m$ | $E_\phi^x$ | $M$ | $S_\alpha$ | $F_\beta^\omega$ | $F_\beta^m$ | $E_\phi^m$ | $E_\phi^x$ | $M$ | $S_\alpha$ | $F_\beta^\omega$ | $F_\beta^m$ | $E_\phi^m$ | $E_\phi^x$ | $M$ |
| M1 | 0.684 | 0.606 | 0.680 | 0.687 | 0.689 | 0.132 | 0.783 | 0.701 | 0.756 | 0.798 | 0.800 | 0.049 | 0.767 | 0.696 | 0.752 | 0.776 | 0.778 | 0.078 |
| M2 | 0.827 | 0.787 | 0.817 | 0.908 | 0.916 | 0.062 | 0.847 | 0.761 | 0.790 | 0.921 | 0.931 | 0.033 | 0.870 | 0.825 | 0.846 | 0.930 | 0.939 | 0.040 |
| M3 | 0.826 | 0.784 | 0.813 | 0.906 | 0.914 | 0.063 | 0.847 | 0.761 | 0.789 | 0.922 | 0.931 | 0.033 | 0.871 | 0.826 | 0.846 | 0.932 | 0.940 | 0.039 |
| M4 | 0.832 | 0.794 | 0.821 | 0.913 | 0.920 | 0.061 | 0.846 | 0.760 | 0.789 | 0.921 | 0.931 | 0.033 | 0.871 | 0.826 | 0.847 | 0.932 | 0.940 | 0.040 |
| **Layer analysis study in FM** | | | | | | | | | | | | | | | | | | |
| Setting | $S_\alpha$ | $F_\beta^\omega$ | $F_\beta^m$ | $E_\phi^m$ | $E_\phi^x$ | $M$ | $S_\alpha$ | $F_\beta^\omega$ | $F_\beta^m$ | $E_\phi^m$ | $E_\phi^x$ | $M$ | $S_\alpha$ | $F_\beta^\omega$ | $F_\beta^m$ | $E_\phi^m$ | $E_\phi^x$ | $M$ |
| $2-layers$ | 0.824 | 0.783 | 0.813 | 0.906 | 0.914 | 0.064 | 0.846 | 0.760 | 0.789 | 0.922 | 0.932 | 0.032 | 0.870 | 0.827 | 0.849 | 0.931 | 0.939 | 0.040 |
| $4-layers$ | 0.829 | 0.789 | 0.818 | 0.906 | 0.915 | 0.063 | 0.847 | 0.762 | 0.791 | 0.921 | 0.931 | 0.033 | 0.869 | 0.825 | 0.847 | 0.929 | 0.938 | 0.041 |
| $8-layers$ | 0.832 | 0.794 | 0.821 | 0.913 | 0.920 | 0.061 | 0.846 | 0.760 | 0.789 | 0.921 | 0.931 | 0.033 | 0.871 | 0.826 | 0.847 | 0.932 | 0.940 | 0.040 |
| $16-layers$ | 0.830 | 0.791 | 0.821 | 0.909 | 0.917 | 0.064 | 0.845 | 0.761 | 0.791 | 0.921 | 0.931 | 0.033 | 0.869 | 0.826 | 0.850 | 0.929 | 0.938 | 0.041 |
| **Input analysis study in FM** | | | | | | | | | | | | | | | | | | |
| Setting | $S_\alpha$ | $F_\beta^\omega$ | $F_\beta^m$ | $E_\phi^m$ | $E_\phi^x$ | $M$ | $S_\alpha$ | $F_\beta^\omega$ | $F_\beta^m$ | $E_\phi^m$ | $E_\phi^x$ | $M$ | $S_\alpha$ | $F_\beta^\omega$ | $F_\beta^m$ | $E_\phi^m$ | $E_\phi^x$ | $M$ |
| $I+I$ | 0.832 | 0.794 | 0.822 | 0.911 | 0.919 | 0.061 | 0.844 | 0.755 | 0.783 | 0.920 | 0.930 | 0.034 | 0.868 | 0.822 | 0.843 | 0.930 | 0.938 | 0.041 |
| $I+D$ | 0.834 | 0.795 | 0.821 | 0.912 | 0.920 | 0.061 | 0.846 | 0.758 | 0.785 | 0.919 | 0.929 | 0.033 | 0.870 | 0.824 | 0.844 | 0.931 | 0.939 | 0.041 |
| $D+D$ | 0.832 | 0.794 | 0.821 | 0.913 | 0.920 | 0.061 | 0.846 | 0.760 | 0.789 | 0.921 | 0.931 | 0.033 | 0.871 | 0.826 | 0.847 | 0.932 | 0.940 | 0.040 |

**Table 3: Ablation study for each module of the proposed DSAM on COD datasets, layer analysis study in FM and input analysis study in FM. SAM (M1): This model is SAM for segment everything mode. We evaluate the mask with the best segmentation quality in this mode as the final result of segmentation. SAM + PDM (M2) : providing PDM based on M1. SAM + FM (M3): providing FM based on M1. SAM + PDM + FM (M4): providing PDM and PM based on M1. $I+I$ denotes image embedding and image embedding. $I+D$ denotes image embedding and depth embedding. $D+D$ denotes depth embedding and depth embedding.**

**Qualitative Result.** Fig. 4 illustrates the comparison of the proposed method with some representative competitors in various types of scenarios. These comparative maps are derived from three different test datasets. The performance of our DSAM is analyzed in four aspects: target size, edge complexity, ease of missing parts, and multiple targets. For both large targets (the third row) and small targets (the first row), DSAM manages to balance contour and detail effectively. For targets with high edge complexity (the eleventh row), our network can still achieve clear segmentation of the edges. Also, DSAM can correctly segment easily overlooked or highly camouflaged parts (the sixth row). In the case of multiple targets (the ninth and tenth rows), other networks might exhibit either omission or excessive segmentation. However, our model can accurately identify disguised targets and perform appropriate segmentation.

## 4.4 Ablation Study

Ablation experiments are conducted to assess the effectiveness of PDM and FM. PDM and FM are sequentially added to SAM, and the training setup is maintained. Four models are designed to evaluate the efficacy of these modules. The comparison results across CAMO, COD10K, and NC4K datasets are presented in Tab. 3. Each module demonstrates a constructive impact on the experimental results, and our proposed DSAM attains SOTA performance.

**Effectiveness of PDM.** The efficiency of FDM can be demonstrated through a comparison between M1 and M2. Transitioning from M1 to M2 on CAMO, the top three metrics show great improvements: an increase of 18.1% in $F_\beta^\omega$, 22.1% in $E_\phi^m$, and 22.7%

in $E_\phi^x$. On COD10K, apart from MAE, the average improvement in positive evaluation criteria reaches 8.24%. On NC4K, the least performing positive metric ( $F_\beta^m$) exhibits an improvement of 9.4%. From the above data, it can be concluded that PDM enhances the performance of SAM-based models in the COD domain.

**Effectiveness of FM.** The efficiency of FM can be demonstrated by comparing M1 and M3. From M1 to M3, on the CAMO dataset, the average improvement in positive evaluation criteria is 17.94%. On COD10K, the negative indicator, MAE, decreases by 1.6%. On NC4K, the most improved forward indicator ($E_\phi^x$) increases by 16.2%. From the above data, it can be concluded that FM significantly improves performance.

To delve deeper, the rationality of the internal structural design of FM is experimentally analyzed. Firstly, this study analyzes the rationale behind utilizing depth embedding as inputs in FM. The input embedding combination of FM is modified while keeping the other network designs unchanged. The specific experimental data are presented in the sub-table (input analysis study in FM) in Tab. 3. To gain a clearer understanding of the relationship between evaluation metrics and the type of input embedding combination, the table is visualized in Fig. 5. Each subplot in the figure demonstrates that the combination of dual-depth embedding is optimal, so this design is adopted. Then, this study analyzes the rationality of dividing the teacher embedding into 8 segments by channel during the mask reversion operation. While keeping the other network designs of DSAM unchanged, the partitioning layers of the mask reversion operation are changed to 2 layers, 4 layers, 8 layers, and

| Scale analysis of prediction maps study in FM | | | | | | | | | | | | | | | | | |
|---|---|---|---|---|---|---|---|---|---|---|---|---|---|---|---|---|---|
| Ratio | CAMO | | | | | | COD10K | | | | | | NC4K | | | | | |
| | $S_\alpha$ | $F_\beta^\omega$ | $F_\beta^m$ | $E_\phi^m$ | $E_\phi^x$ | $M$ | $S_\alpha$ | $F_\beta^\omega$ | $F_\beta^m$ | $E_\phi^m$ | $E_\phi^x$ | $M$ | $S_\alpha$ | $F_\beta^\omega$ | $F_\beta^m$ | $E_\phi^m$ | $E_\phi^x$ | $M$ |
| 3 : 7 | 0.823 | 0.782 | 0.813 | 0.905 | 0.914 | 0.066 | 0.846 | 0.763 | 0.792 | 0.923 | 0.933 | 0.033 | 0.870 | 0.827 | 0.849 | 0.931 | 0.940 | 0.040 |
| 2 : 8 | 0.828 | 0.788 | 0.817 | 0.904 | 0.912 | 0.063 | 0.845 | 0.758 | 0.787 | 0.920 | 0.929 | 0.033 | 0.869 | 0.825 | 0.847 | 0.930 | 0.939 | 0.041 |
| 1 : 9 | 0.832 | 0.794 | 0.821 | 0.913 | 0.920 | 0.061 | 0.846 | 0.760 | 0.789 | 0.921 | 0.931 | 0.033 | 0.871 | 0.826 | 0.847 | 0.932 | 0.940 | 0.040 |
| 0.5 : 9.5 | 0.832 | 0.795 | 0.823 | 0.911 | 0.919 | 0.061 | 0.846 | 0.758 | 0.786 | 0.920 | 0.930 | 0.033 | 0.869 | 0.823 | 0.844 | 0.930 | 0.938 | 0.041 |
| Loss ratio analysis | | | | | | | | | | | | | | | | | |
| Ratio | CAMO | | | | | | COD10K | | | | | | NC4K | | | | | |
| | $S_\alpha$ | $F_\beta^\omega$ | $F_\beta^m$ | $E_\phi^m$ | $E_\phi^x$ | $M$ | $S_\alpha$ | $F_\beta^\omega$ | $F_\beta^m$ | $E_\phi^m$ | $E_\phi^x$ | $M$ | $S_\alpha$ | $F_\beta^\omega$ | $F_\beta^m$ | $E_\phi^m$ | $E_\phi^x$ | $M$ |
| 3 : 7 | 0.831 | 0.796 | 0.829 | 0.908 | 0.918 | 0.062 | 0.847 | 0.763 | 0.794 | 0.922 | 0.932 | 0.032 | 0.869 | 0.826 | 0.850 | 0.929 | 0.939 | 0.041 |
| 2 : 8 | 0.831 | 0.791 | 0.820 | 0.908 | 0.916 | 0.061 | 0.847 | 0.760 | 0.788 | 0.924 | 0.934 | 0.033 | 0.870 | 0.824 | 0.845 | 0.931 | 0.939 | 0.041 |
| 1 : 9 | 0.832 | 0.794 | 0.821 | 0.913 | 0.920 | 0.061 | 0.846 | 0.760 | 0.789 | 0.921 | 0.931 | 0.033 | 0.871 | 0.826 | 0.847 | 0.932 | 0.940 | 0.040 |
| 0.5 : 9.5 | 0.832 | 0.792 | 0.819 | 0.911 | 0.918 | 0.062 | 0.843 | 0.754 | 0.783 | 0.918 | 0.927 | 0.034 | 0.866 | 0.819 | 0.842 | 0.927 | 0.935 | 0.043 |

**Table 4: Hyper-parameter sensitivity analysis. In the Scale analysis table, the ratio is set to $Pred_{FM}$: $Pred_{SAM}$. In the Loss ratio table, the ratio is set to $Loss_{PDM}$:$Loss_{SAM}$**

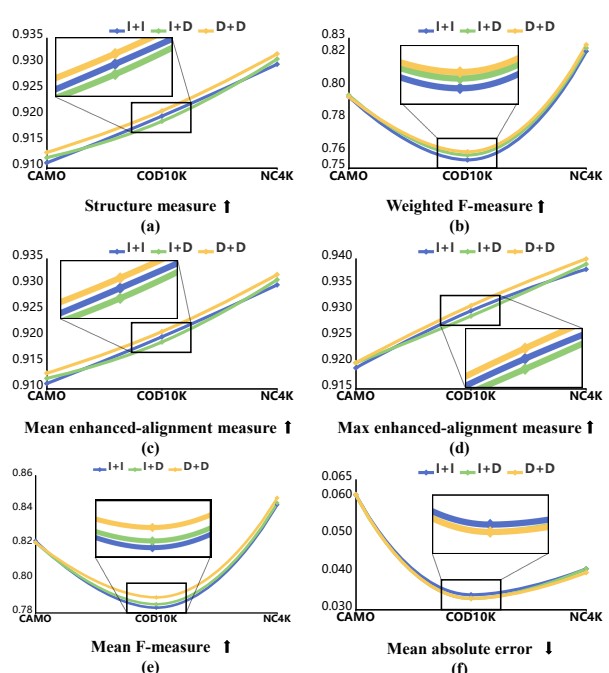

**Figure 5: Ablation experiments on the input to the FM. ↑ indicates the higher the score the better and ↓ indicates the lower the score the better. $I$ stands for image embedding acting as student embedding, and $D$ stands for depth embedding acting as teacher embedding.**

16 layers, respectively. The specific experimental data is shown in the sub-table (layer analysis study in FM) in Tab. 3 and visualized in Fig. 6. In Fig. 6, when the number of layers is 8, the average positive evaluation criterion is highest, and the negative evaluation criterion is lowest, demonstrating that 8 layers represent the optimal solution.

**Setting of Hyper-parameters.** Two ablation experiments are conducted to investigate the proportion between the maps predicted by FM and those predicted by the original SAM, as well as

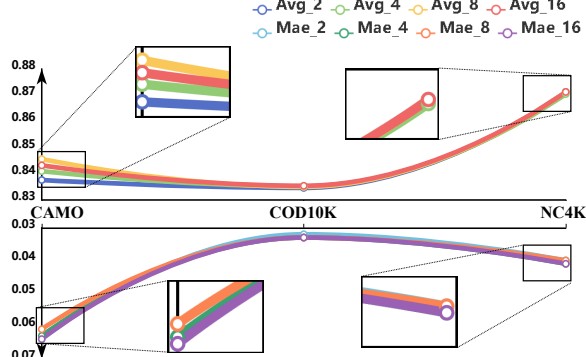

**Figure 6: Ablation experiments on the layer number to the FM. $Avg\_x$ represents the average score of positive indicators when the number of layers is $x$. $Mae\_x$ represents the $Mae$ when the number of layers is $x$.**

the corresponding relationship between the original loss function of SAM and the distillation loss function of PDM. The specific experimental data are shown in Tab. 4. From the sub-tables (scale analysis of prediction maps in FM, and loss ratio analysis), it can be observed that, in both cases, the data decline towards both sides from the third group, indicating that 1 : 9 is the optimal ratio.

## 5 CONCLUSION

This paper proposes DSAM, a novel network architecture integrating SAM into COD by fully leveraging the complementarity among multiple modalities. In DSAM, PDM and FM are proposed to explore and utilize depth information, providing a novel way for exploiting the interaction between RGB and depth data. The state-of-the-art performance of DSAM is demonstrated over 17 cutting-edge methods across multiple datasets. However, DSAM exhibits internal over-segmentation of the target, wherein target edges are accurately segmented but internal regions suffer from erroneous voids. Our future research will attempt to overcome this challenge through the integration of diffusion methodologies.

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
