# OpenReview forum: "Exploring Deeper! Segment Anything Model with Depth Perception for Camouflaged Object Detection"
_acmmm.org/ACMMM/2024/Conference — MM2024 Poster_

### Official Review · Reviewer_AtjZ · 2024-05-24

**Rating:** 2
**Confidence:** 2

**Summary:**

To enhance the effectiveness of SAM in camouflaged object detection, this article incorporates depth information onto the basis of SAM. By employing knowledge distillation and Bias Correction methods, it achieves the interaction between RGB features and depth features. Through the complementarity of RGB and depth information, detection performance is improved, leading to SOTA performance in extensive comparative experiments.

**Strengths:**

1.  This article introduces a new form of prompt by combining box prompt with depth information.
2.  This article extends the applicability of the SAM model to disguised scenes by incorporating depth information.

**Limitations:**

1. Based on the experimental results in the article, I have the following questions:
1) From my knowledge, the HitNet model performs well in camouflage object detection. However, I do not see any comparison with it in Table 1. Why is that?
2) Looking at the comparison between FSPNet and Ours in Table 1, the performance of both models is quite similar. Does depth information really make a difference? Additionally, according to the results in the bottom table of Table 3, when comparing I+D with I+I, most metrics show only a marginal improvement of 0.001-0.003, and some metrics even decrease, why does using D+D configuration yield the best results?
2. The abstract of the article mentions the interaction between RGB and depth features. I have a question: Is this approach effective? There are no ablation experiments regarding this aspect.
3. In Section 3.2 of the article, it is mentioned that the depth map is not obtained through a depth sensor. So, is it generated? If it is generated, is this depth valid?
4. There are some formatting issues in the article, such as the titles of tables should be placed on top. There are also some flaws in writing, for example, on line 137, "as demonstrated in the left part of Fig. 1" should be "Fig. 2" instead of "Fig. 1".

**Suitability:**

3

---

### Official Review · Reviewer_CEri · 2024-06-09

**Rating:** 4
**Confidence:** 2

**Summary:**

This paper introduces the DSAM method for camouflaged object detection, where depth information is utilized to enhance the segmentation performance. Specifically, a Prompt-Deeper Module (PDM) and a Finer Module (FM) are proposed for extracting depth-aware box prompt and refining the segmentation results by identifying missed depth cues. DSAM achieves SOTA results on various COD benchmarks.

**Strengths:**

1. This paper proposes to enhance the segmentation results of SAM for COD with depth information, which is reasonable and sound.
2. The analysis and experiments are comprehensive.
3. Compared to SOTA methods, DSAM uses less training resources and is more efficient.

**Limitations:**

1. This method needs to forward SAM for several times (e.g. in the image encoder and in the FM), how is the inference time compared to other methods? It is better to analyze the running time of different modules in DSAM.
2. This paper claims (in L346) that using knowledge distillation instead of direct fuse depth map will reduce the noise effect. It is better that this claim can be further elaborated or analyzed with evidence in the paper.

**Suitability:**

3

---

### Official Review · Reviewer_LgRG · 2024-06-09

**Rating:** 4
**Confidence:** 2

**Summary:**

This paper proposes a new Segment Anything Model with Depth Perception for Camouflaged Object Detection It exploits the zero-shot capability of  SAM to obtain precise segmentation in RGBD domain. It utilizes PDM and FM to achieve the interaction between RGB and depth features. Then the authors integrate some designed modules to improve the performance. The experimental results show the good performance of this paper.

**Strengths:**

1. The writing of this paper is relatively good, so the reviewer is easy to follow the idea.
2. The results of this paper is good on two public datasets.
3. The experiment validation is relatively strong and rich, which enhances the persuasiveness of this paper.

**Limitations:**

1. The author should provide some description of the metrics about their expected trend (↑or↓).

2. In this paper, the depth information is obtained by PDM, but there are many other methods that can give the depth information. If PDM is replaced with other depth-perception methods, can this method still work? Can the author provide some analysis of this ?

3. Please compare with these key references:
[1] High-Resolution Iterative Feedback Network for Camouflaged Object Detection
[2] A systematic review of image-level camouflaged object detection with deep learning
[3]Relax Image-Specific Prompt Requirement in SAM: A Single Generic Prompt for Segmenting Camouflaged Objects，

4. Please make Figure 2 more explanatory by adding annotations, especially on the left side of the figure, and briefly describe the model process.
5. Please provide a brief description of the process in Figure 3, rather than just explaining the meaning of the annotations.
6. Figure 4: Please add a result graph of DSAM and briefly describe the reason for its generation..
7. Can the results of the manuscript be reproduced based on the details provided in the methods section? In order to make the results reproducible, more information should be added.
8. A better conclusion statement is needed, as your conclusion is limited to a brief statement without evidence to support it.

**Suitability:**

2

---

### Meta-Review · Area_Chair_C1Jo · 2024-06-24

**Recommendation:** Accept (Poster)
**Confidence:** 5

**Metareview:**

The submission initially received two borderline accepts and one weak reject. Given the mixed ratings, the Area Chair (AC) reviewed the paper further and decided to accept it. The authors are advised to integrate the comparison results from the rebuttal into the camera-ready version.